# Triel Bond Formed by Malondialdehyde and Its Influence on the Intramolecular H-Bond and Proton Transfer

**DOI:** 10.3390/molecules27186091

**Published:** 2022-09-18

**Authors:** Qiaozhuo Wu, Shubin Yang, Qingzhong Li

**Affiliations:** The Laboratory of Theoretical and Computational Chemistry, School of Chemistry and Chemical Engineering, Yantai University, Yantai 264005, China

**Keywords:** triel bond, hydrogen bonding, proton transfer, NBO

## Abstract

Malondialdehyde (MDA) engages in a triel bond (TrB) with TrX_3_ (Tr = B and Al; X = H, F, Cl, and Br) in three modes, in which the hydroxyl O, carbonyl O, and central carbon atoms of MDA act as the electron donors, respectively. A H···X secondary interaction coexists with the TrB in the former two types of complexes. The carbonyl O forms a stronger TrB than the hydroxyl O, and both of them are better electron donors than the central carbon atom. The TrB formed by the hydroxyl O enhances the intramolecular H-bond in MDA and thus promotes proton transfer in MDA-BX_3_ (X = Cl and Br) and MDA-AlX_3_ (X = halogen), while a weakening H-bond and the inhibition of proton transfer are caused by the TrB formed by the carbonyl O. The TrB formed by the central carbon atom imposes little influence on the H-bond. The BH_2_ substitution on the central C-H bond can also realise the proton transfer in the triel-bonded complexes between the hydroxyl O and TrH_3_ (Tr = B and Al).

## 1. Introduction

Malondialdehyde (MDA), a naturally occurring product in lipid peroxidation and prostaglandin biosynthesis [1], has been known as a biomarker of lipid oxidation induced by reactive oxygen species [2], a reliable biomarker for bipolar disorder [3], or an oxidative stress marker in oral squamous cell carcinoma [4]; thus, it has received much attention [5,6,7,8,9,10,11,12,13]. This molecule exhibits an intramolecular proton transfer with two equivalent forms separated by a barrier of medium height [5]. The intramolecular proton transfer in MDA involves a 2.2 kcal/mol lower barrier than that in its radical analogues [6]. When a NO_2_ or BH_2_ group is attached to the central carbon atom, the barrier is reduced to less than 1 kcal/mol [7]. Theoretical and experimental studies showed that 2-chloromalonaldehyde exhibits a weaker intramolecular H-bond than MDA [8,9]. However, the intramolecular H-bond in MDA is strengthened if strong electron donors and/or sterically hindered substituents are present in its two side carbon atoms [7]. The IR and UV spectra of MDA were also measured in the gas phase and water [10,11]. The MDA in water has a slightly red-shifted UV spectrum compared with that in the gas phase [10]. Tunnelling occurs in MDA, and its mechanisms can be understood through the isotope effect (IE), which is classified into primary IE and secondary IE [12]. The primary H/D kinetic IE on the intramolecular proton transfer in MDA is dominated by zero-point energy effects, and tunnelling plays a minor role at room temperature [13].

Intra- or intermolecular proton transfer reactions are considered to be one of the most fundamental and important processes in chemistry and biology, such as acid–base neutralisation reactions and enzymatic reactions [14]. The proton transfer in MDA can be regulated through cooperativity with other interactions. When a BeH_2_ or BeF_2_ group engages in a beryllium bond with the hydroxyl/carbonyl group of MDA, the intramolecular H-bond in MDA is strengthened or weakened [15], which is accompanied by the inhibition/promotion of proton transfer. This effect is also realised by adding F_2_SiO to MDA, where a tetrel bond is formed [16]. In general, the stronger the interaction imposed, the more prominent the effect. The stronger interaction makes the binding distance of the weaker one undergo a larger change.

A triel bond (TrB) is an attractive interaction that occurs between the triel atom such as B or Al and an electron donor [17]. Such interactions are usually so strong that they have many of the characteristics of covalent bonds and can even be classified as typical covalent bonds [18]. Triel atoms are usually sp^2^-hybridised with a π–hole above and below the molecular plane. Interestingly, when the sp^2^-hybridised triel atom binds to a strong Lewis base, it may become sp^3^-hybridised [19]. Thus, the trivalent centres in those complexes with such strong interactions follow the octet rule and can be classified as tetravalent centres, which usually have a tetrahedral structure, indicating a large geometric deformation of the interacting species, a feature of TrB formation. For example, an isolated BH_3_ monomer has a planar triangular structure, while the BH_3_∙∙∙NH_3_ complex is tetrahedral [20]. This bond plays an important role in energy materials, chemical reactions, and biological systems [21,22,23,24]. For instance, the strong organoborane Lewis acid B(C_6_F_5_)_3_ catalyses the hydrosilation of aromatic and aliphatic carbonyl functions at convenient rates, with loadings of 1−4% [22]. TrB also contributes to the molecular hydrogen release process, which has also been explored as a research topic related to hydrogen storage materials [23]. Thus, TrB has garnered more attention in recent years [25,26,27,28,29,30,31,32,33,34,35]. Although different types of electron donors are utilised in the TrBs [32,33,34,35], the most common ones are from molecules with lone pairs such as N and O. A comparison was made for the TrBs with different chalcogen electron donors (H_2_O, H_2_S, and H_2_Se), and it was found that H_2_O forms a stronger TrB [36]. This bond displays some different properties from H-bond. In HCN∙∙∙HCN-BF_3_, where a TrB and H-bond coexist, the strong TrB suffers a larger shortening than the weaker H-bond [37]. Thus, it is interesting to study the influence of TrB on the intramolecular H-bond in MDA and its proton transfer.

In this article, the complexes of MDA and TrX_3_ (Tr = B and Al; X = H, F, Cl, and Br) are studied. There are two types of oxygen atoms (hydroxyl and carbonyl O atoms) in MDA; thus, their ability to bind with a triel atom is compared. In addition, the central carbon atom has negative molecular electrostatic potentials (MEPs), as shown in Figure 1. Therefore, this negative MEP region also binds with TrX_3_, resulting in a π–π TrB. On the other hand, we focus on the influence of TrB on the intramolecular H-bond in MDA and the corresponding proton transfer. When a BH_2_ group is attached to the central carbon atom, the barrier in the proton transfer reduces [7]. Thus, a combination of substitution and cooperative effects is used to regulate the proton transfer.

## 2. Results

### 2.1. Coplanar Triel-Bonded Complexes

Malondialdehyde (MDA) is often used as a model for studying intramolecular proton transfer, and TrX_3_ can be added to MDA to form a TrB that in turn affects the proton transfer. As can be seen from the MEP diagram of MDA in Figure 1, there are both positive red and negative blue regions in MDA, indicating that it can act as both an electron donor and an acceptor. MDA contains two blue regions, which are located, respectively, at the hydroxyl oxygen and the carbonyl oxygen. It is clear that the latter has a more negative MEP value than the former. The three CH hydrogen atoms have red areas. Due to the delocalisation of the ring structure of MDA, the three carbon atoms have negative MEPs; thus, they can also bind with the π–hole of TrX_3_, which has been studied in previous studies (Appendix A).

The π–hole on the T atom of TrX_3_ may interact with the hydroxyl O(1) or carbonyl O(2) atom of MDA to form a TrB, respectively, designated by the “a” and “b” labels. Figure 2 and Figure 3 show the structures of the a-type and b-type complexes, respectively. The Tr∙∙∙O distance is shorter than 2 Å in most complexes, with an exception in MDA-BF_3_-a. The shorter Tr∙∙∙O distance means that the TrB formed between both molecules is very strong. The halogen substitution shortens the Tr∙∙∙O distance relative to the TrH_3_ complex in most cases. Only BF_3_ elongates the Tr∙∙∙O distance in spite of the largest π–hole on the B atom.

Since Figure 1 confirms that the carbonyl O of MDA has a more negative MEP value than the hydroxyl O, it is not surprising that the former forms a stronger TrB with TrX_3_ than the latter. This difference is reflected in the shorter Tr∙∙∙O distance in the a-configuration, the more negative interaction energy (Table 1), and the greater electron density at the Tr∙∙∙O BCP (Table 2). However, there are exceptions for the structures of MDA-BCl_3_, MDA-BBr_3_, MDA-AlF_3_, MDA-AlCl_3_, and MDA-AlBr_3_ since both configurations have the same interaction energy, although they are formed in very different ways. As shown in Table 1, the interaction energy E_int_ in most of the MDA-AlX_3_ complexes is larger than that of that in the MDA-BX_3_ analogue, which may be due to the fact that the π–hole values of AlX_3_ are larger than those of BX_3_. When the H atoms of TrH_3_ are replaced by halogen, E_int_ basically increases, except for MDA-BF_3_-a, which also has the lowest E_int_ in all the complexes, although its π–hole is not the smallest. With the increase in halogen electronegativity, E_int_ increases for the MDA-AlX_3_ complexes but decreases for the MDA-BX_3_ complexes. This inconsistent change is mainly attributed to the distortion of TrX_3_. BF_3_ shows a smaller distortion relative to BCl_3_ and BBr_3_ in the complex; thus, an abnormal change occurs for the BF_3_ complexes.

The binding energy E_b_ and deformation energy DE of these complexes are also given in the last two columns of Table 1. The binding energy is the difference between the energy of the complex relative to the sum of the energies of the isolated monomers (in their optimised geometry). In general, E_b_ has the same trend as E_int_, but E_b_ is not as negative as E_int_, and the difference between them is DE. The DE values in MDA-BCl_3_ and MDA-BBr_3_ are large enough (>24 kcal/mol), which is about double as much as E_b_. This is also taken as a character of a TrB since the triel molecule is often distorted easily. This distortion easily occurs in the MDA-BX_3_ (X = H, Cl, and Br) complexes, with a DE as much as double that in the MDA-AlX_3_ analogues.

The formation of a TrB can be further confirmed by a complicated colour region in the NCI analysis (Appendix A). In most complexes, this colour region between the Tr and O atoms is overlapped with red and blue; thus, the TrB is very strong. In addition, a green region is found between one Tr-X bond of TrX_3_ and the C-H bond adjoined with the O atom that binds with TrX_3_, corresponding to a weak H-bond with a long H∙∙∙X distance. Both MDA and TrX_3_ play a reverse role in both the TrB and H-bond; thus, both interactions display positive cooperativity with each other.

Table 2 collects the AIM data of the Tr∙∙∙O triel bond, including the important parameters of the electron density (ρ), its Laplacian (∇^2^ρ), and the energy density (H) at the bond critical point (BCP) of the TrB. For the MDA-BX_3_ complex, ∇^2^ρ is positive but H is negative, indicating that the TrB is partially covalent interaction, which is also confirmed by the magnitude of ׀V׀/G (>1). However, for the MDA-AlX_3_ complex, both ∇^2^ρ and H are positive, and the magnitude of ׀V׀/G is smaller than 1, suggesting that this TrB is a completely closed shell interaction [38], which is inconsistent with its relatively strong interaction energy. Thus, the estimation of the nature of the TrB according to the topological parameters should be made with caution. In each type of complex, a good linear relationship is present between the electron density and the interaction energy for the B∙∙∙O TrB, while an opposite dependence is found for the Al∙∙∙O TrB. This again highlights the careful consideration necessary in studying a TrB according to the AIM parameters.

The charge transfer (CT) values for the different types of binary complexes are given in Appendix A. In most complexes, CT is larger than 0.1 e, indicative of a stronger TrB. A good relationship is not found between CT and E_int_, partly due to the coexistence of both the TrB and the H∙∙∙X H-bond with an opposite direction of CT.

To better understand the nature of the Tr∙∙∙O TrB, an energy decomposition analysis was carried out for these systems. As shown in Appendix A, the interaction energy was decomposed into five terms, including the electrostatic energy (E^ele^), the exchange energy (E^ex^), the repulsion energy (E^rep^), the polarisation energy (E^pol^), and the dispersion energy (E^disp^). In most cases, E^ex^ is the largest negative term, but it is usually cancelled with E^rep^; thus, neither of these two terms is discussed. E^ele^ is larger than E^pol^ and E^disp^, and E^pol^ is comparable with E^ele^ in most cases, which supports the conclusion that the TrB is very strong. E^disp^ is negative in the B∙∙∙O TrB but becomes positive in the Al analogue due to the very shorter Al∙∙∙O distance.

### 2.2. π–π Parallel Structures

A closer look at the MEP of the MDA molecule in Figure 1 shows that, in addition to the negatively charged blue region at the hydroxyl O-terminus and carbonyl O-terminus, there is also a relatively small negatively charged region above the C-atom at the centre of the MDA molecular plane, so when the π–hole on the Tr atom of TrX_3_ approaches the central carbon atom of MDA from above, a face-to-face parallel π–π structure appears, as shown in Figure 4, like the π–π interactions in the aromatic systems.

The π–π structure involving BH_3_ has comparable stability with the a-type analogue but is much less stable for its halogen derivatives. AlX_3_ also forms a weaker π–π structure with MDA again with the a-type analogue. The halogen substitution of AlX_3_ has a similar enhancing effect on the stability of the π–π structure, while an opposite influence is found for the halogen substitution of BX_3_. The effect of different halogen atoms has a small difference. Likely, AlX_3_ engages in a stronger π–π TrB than BX_3_ except for X = H. The above conclusions are obtained according to the interaction energy in Table 1 and the Tr∙∙∙C distance in Figure 4.

The distribution of the NCI region between TrX_3_ and MDA in the π–π structure (Appendix A) is like that in the π–π stacking of two benzene molecules [39]. The NCI region in the π–π structure of AlX_3_ has a deeper and more complicated colour, consistent with a stronger TrB. The electron density at the Tr∙∙∙C BCP does not reflect the change in the interaction energy in the π–π structure since this structure is not bounded only by a Tr∙∙∙C TrB. This is also true for the Laplacian at the Tr∙∙∙C BCP. The sign of the energy density is negative for the Tr∙∙∙C BCP with the interaction energy larger than 10 kcal/mol.

Although MDA-BH_3_-c has the smaller interaction energy, CT in MDA-BH_3_-c is larger than that in MDA-BH_3_-a and MDA-BH_3_-b since the π electrons in the ring of MDA are easily lost.

If the interaction energy is smaller than 5 kcal/mol, the polarisation contribution is smallest, and the dispersion is even larger than the electrostatic energy in MDA-BCl_3_-c and MDA-BBr_3_-c due to the nature of π electron in the MDA ring and the longer distance (>3 Å). If the interaction energy is larger than 10 kcal/mol, the dispersion contribution is the smallest, and the polarisation energy is comparable with the electrostatic energy.

### 2.3. Proton Transfer

The formation of the TrB has an important influence on the intramolecular structures of MDA, particularly its intramolecular H-bond. When the Tr atom of TrX_3_ participates in a TrB with the hydroxyl O of MDA, R_2_(O-H) is stretched, and R_1_(H∙∙∙O) is shortened. In MDA-AlF_3_-a, MDA-BCl_3_, and MDA-BBr_3_, R_1_(H∙∙∙O) is much shorter than R_2_(O-H), which can be described as a partial proton transfer. However, when the Tr atom of TrX_3_ engages in a TrB with the carbonyl O of MDA, R_2_(O-H) is shortened, and the R_1_(H∙∙∙O) is stretched, indicating that no proton transfer occurs. When the Tr atom of TrX_3_ forms a π–π parallel structure with the central C atom of MDA, most of the structures have a small degree of R_2_(O-H) elongation and a shortened R_1_(H∙∙∙O), which can also be considered to have undergone proton transfer but to a much lesser extent than the a-configuration. It is also interesting to note that the R_2_ (O-H) and R_1_(H∙∙∙O) in both MDA-BH_3_-c and MDA-BF_3_-c are slightly elongated, a change that is negligible. The above bond lengths as well as their bond length variations are listed in Table 3.

Table 4 shows the AIM analysis for the intramolecular H···O(2) and O(1)-H BCPs. In the MDA monomer, both ∇^2^ρ and H at the O(1)-H BCP are negative, with a character of a covalent bond, while only H is negative at the H···O(2) BCP, indicative of a partially covalent interaction. When the Tr atom of TrX_3_ forms a TrB with the hydroxyl O of MDA, the ρ at the H∙∙∙O(2) BCP increases, while it is decreased for the O(1)-H BCP. Even the sign of ∇^2^ρ at both types of BCPs is changed in MDA-BX_3_-a (X = Cl and Br) and MDA-AlX_3_-a (X = F, Cl, and Br). Specifically, ∇^2^ρ becomes negative for the H∙∙∙O(2) BCP but positive for the O(1)-H BCP. In addition, H at the H∙∙∙O(2) BCP is more negative but less negative for the O(1)-H BCP. These changes demonstrate that the intramolecular H-bond is strengthened, and even a proton transfer occurs in the a-type complex with a very strong TrB. This enhancing effect is also found in the c-type complex except for MDA-BH_3_-c and MDA-BF_3_-c, but no proton charge occurs. When the Tr atom in TrX_3_ forms a TrB with the carbonyl O of MDA, an opposite change is found for the ρ at the H···O(2) and O(1)-H BCPs, and the signs of both ∇^2^ρ and H are not changed. This means that the intramolecular H-bond is weakened.

### 2.4. Substitution Effect

Besides external influences, the intramolecular H-bond of MDA can also be regulated by the presence of substituents. In order to enhance the H-bond within the MDA molecule, we selected the three structures of MDA-BH_3_-a, MDA-BF_3_-a, and MDA-AlH_3_-a since no complete proton transfer occurs and replaces the H atom attached to the central C atom in these structures with a BH_2_ group, as shown in Figure 5. It is clear from the figure that the structures of BH_2_-MDA-BH_3_-a and BH_2_-MDA-AlH_3_-a dramatically change, compared with the corresponding MDA-BH_3_-a and MDA-AlH_3_-a, while the structure of BH_2_-MDA-BF_3_-a has a slight change.

From Table 3, it can be seen that there is a large degree of contraction of R_1_ and a significant degree of stretching of R_2_ in both BH_2_-MDA-BH_3_-a and BH_2_-MDA-AlH_3_-a, compared with MDA-BH_3_-a and MDA-AlH_3_-a, which could indicate a significant enhancement in the H-bond and hence proton transfer. This conclusion is also deduced from the AIM analysis in Table 4, which shows that the ρ at the H∙∙∙O BCP in both the BH_2_-MDA-BH_3_-a and BH_2_-MDA-AlH_3_-a structures is considerably increased, and the sign of ∇^2^ρ changes from positive in MDA-BH_3_-a and MDA-AlH_3_-a to negative, indicating that the H∙∙∙O H-bond in these structures has changed from the partially covalent interaction in MDA-BH_3_-a and MDA-AlH_3_-a to the present covalent bond. In contrast, the ρ at the O-H BCP in both BH_2_-MDA-BH_3_-a and BH_2_-MDA-AlH_3_-a decreases, and the sign of ∇^2^ρ changes from negative to positive, indicating that the O-H changes in these structures from a covalent bond to a partially covalent interaction. However, when the H atom attached to the C atom in the centre in MDA-BF_3_-a is replaced by a BH_2_ group, there is only a small contraction of R_1_ and a slight increase in ρ at the H∙∙∙O BCP, so we believe that the effect on the intramolecular H-bond is still very small in BH_2_-MDA-BF_3_-a, like MDA-BF_3_-a.

The orbital interaction diagram also demonstrates that the addition of the BH_2_ moiety enhances the H-bond and further promotes proton transfer. In MDA-BH_3_-a (Figure 6a), the charge density moves from the lone pair orbital of the carbonyl O(2) atom into the antibonding orbital of the O(1)-H bond, while this orbital interaction becomes LP_O(1)_→σ*_O(2)-H_ in BH_2_-MDA-BH_3_-a (Figure 6c); thus, the direction of the orbital interaction is reversed. In addition, the overlapping between both orbitals is almost the same in both structures. This similar orbital interaction is also found in MDA-BCl_3_-a (Figure 6b).

The addition of an electron-withdrawing group BH_2_ to the central C atom of MDA also influences the strengths of the TrB and secondary H∙∙∙X interactions. The Tr∙∙∙O and H∙∙∙X distances are shortened in BH_2_-MDA-BH_3_-a and BH_2_-MDA-AlH_3_-a but elongated in BH_2_-MDA-BF_3_-a. Thus, both types of interactions are strengthened in BH_2_-MDA-BH_3_-a and BH_2_-MDA-AlH_3_-a but weakened in BH_2_-MDA-BF_3_-a. In turn, the change in TrB strength would impose an influence on the proton transfer. Therefore, the introduction of the BH_2_ group can regulate the intramolecular proton transfer not only through a substitution effect but also through a cooperative effect.

## 3. Discussion

There is a hydroxyl O in MDA, which can form a TrB with TrX_3_ with the interaction energy of 7–42 kcal/mol. The interaction energy exceeds 37 kcal/mol in MDA-BX_3_-a (X = Cl and Br) and MDA-AlX_3_-a (X = halogen) since the hydroxyl O becomes the carbonyl O upon the formation of a TrB. The carbonyl O has a larger negative MEP than the hydroxyl O; thus, the former participates in a stronger TrB. This supports the fact that the carbonyl O is a stronger electron donor in intermolecular interactions such as H-bond [40]. The intramolecular H-bond belongs to a resonance-assisted H-bond, and the charge density on the hydroxyl O is delocalised and thus reduced, which can be used to explain why the TrB formed by the hydroxyl O of MDA is weaker than that of H_2_O (>20 kcal/mol) [36].

A comparison of the interaction energy between MDA-BH_3_-a and MDA-BF_3_-a shows that the F substituents on the B centre weaken the TrB, although the π–hole in the BF_3_ molecule is larger than that in BH_3_. This conclusion has been confirmed in previous studies [25,26,36]. This abnormality is primarily attributed to the back-bonding effect from the F atom into the outer p orbital of the boron atom, and this effect makes the BF_3_ molecule not easily undergo distortion. DE amounts to 50% of the binding energy in MDA-BF_3_-a and 84% in MDA-BH_3_-a. Accordingly, the distortion has a prominent contribution to the interaction energy of TrB. When the carbonyl O binds with BX_3_, the DE percentage is much larger due to the formation of a stronger TrB. MDA-BF_3_-b has a larger DE contribution to the interaction energy than MDA-BH_3_-b, and the variation in the interaction energy becomes normal in both complexes, consistent with the π–hole on the B atom. Whether the a-type or b-type complexes, the DE contribution to the binding energy in the AlX_3_ complex is much smaller than that in the BX_3_ analogue.

When the hydroxyl O of MDA engages in a very strong TrB in MDA-BX_3_-a (X = Cl and Br) and MDA-AlX_3_-a (X = halogen), the corresponding interaction energy is the same as that in the b-type analogue. This is also reflected in the related data of the geometries, AIM, and NBO. This shows that the a-type and the b-type structures of these complexes are the same. This transformation is described with MDA-BCl_3_ as an example (Figure 7). Structure 1 is the initial complex before optimisation; thus, the geometrical parameters in 1 are the same as those in the isolated MDA. As the B∙∙∙O distance is shorter in 2, the O(1)-H bond is elongated, and the H∙∙∙O(2) distance is shortened. Structure 3 has an equivalent distance of the H atom with the two O atoms. Structure 4 is an enantiomer with 2. When the triel bond is strong enough in structure 5, the proton moves completely from the O(1) to O(2), and their roles are reversed.

The intramolecular H-bond is strengthened if TrX_3_ attacks the hydroxyl O of MDA, while an opposite effect is obtained if it is introduced into the carbonyl O. The former effect promotes the proton transfer, while the latter inhibits the proton transfer. Such an effect has also been imposed by introducing a beryllium bond [15] or a tetrel bond [16]. When a strong tetrel bond formed with F_2_SiO (>42 kcal/mol) is introduced to the hydroxyl O of MDA, the ratio of R_1_/R_2_ is 0.57, which is almost equal to that caused by a stronger TrB (37–42 kcal/mol). This shows that the enhancement in the added interaction has a slight effect on the degree of proton transfer if its interaction energy exceeds a threshold value, and here, it may be 37 kcal/mol.

The π–π structure of the c-type complex between TrX_3_ and MDA is very interesting. If the MDA is replaced by benzene, a similar π–π structure has been reported [41]. The strength of the corresponding TrB is also equal for the complexes of TrX_3_ with both MDA and benzene except BH_3_. The effect of halogen substitution on the triel donor is also the same. Specifically, the halogen substitution on the B centre weakens the π–π structure, while that on the Al centre has an opposite effect. If TrX_3_ is changed into F_2_TO (T = C and Si), the π–π structure obtained with MDA has an interaction energy value of 2.5 kcal/mol for F_2_CO and 27 kcal/mol for F_2_SiO [16]. The corresponding interaction energy is also in a similar range in the TrB formed by TrX_3_.

The DE value in MDA-BBr_3_-c is very small (0.2 kcal/mol); thus, the planar structures of both MDA and BBr_3_ are held in the complex, resulting in a π–π structure. When the B∙∙∙C distance in MDA-BBr_3_-c is shortened to 1.8 Å (its structure is shown in Appendix A), the planar structures of both molecules are distorted with a high deformation energy value of 32 kcal/mol, and the corresponding π–π structure disappears. The interaction energy amounts to 33.64 kcal/mol for the distorted structure of MDA-BBr_3_-c. Interestingly, its binding energy is very small (<2 kcal/mol), which is smaller than that in the corresponding π–π structure (~4 kcal/mol). We plotted the energy curve of MDA-BBr_3_-c by changing the B∙∙∙C distance from 1.5 to 3.5 Å (Appendix A). Two minima are found on the potential energy surface, corresponding to the structures in Figure 4 and Appendix A. The distorted structure is more stable than the π–π structure, and the barrier between both structures is 2 kcal/mol.

## 4. Conclusions

The π–hole above TrX_3_ (Tr = B and Al; X = H, F, Cl, and Br) can form a TrB with the hydroxyl O, carbonyl O, and the central carbon atoms of MDA, marked with a, b, and c, respectively. Other than the TrB, there is also a H∙∙∙X secondary interaction in both the a-type complex and the b-type complex. As expected, the carbonyl O engages in a stronger TrB than the hydroxyl O, and both types of O atoms are better nucleophiles than the central carbon atom. When TrX_3_ is introduced into the hydroxyl O in MDA-BX_3_-a (X = Cl and Br) and MDA-AlX_3_-a (X = halogen), the triel-bonded complexes formed have equal stability to the corresponding b-type complex, with a high degree of interaction energy (>37 kcal/mol). The halogen substitution in the triel donor has an enhancing effect on the strength of TrB with an exception in MDA-BF_3_-a and MDA-BX_3_-c. For each type of complex, AlX_3_ shows a higher affinity to MDA than BX_3_ except X = H.

The formation of TrB between TrX_3_ and MDA has an effect on the strength of the intramolecular H-bond in MDA. When TrX_3_ attacks the hydroxyl or carbonyl O atom of MDA, the former interaction strengthens the intramolecular H-bond, while the latter leads to a weakening H-bond. The π–TrB in the c-type complex also has an enhancing effect on the intramolecular H-bond except for MDA-BH_3_-c and MDA-BF_3_-c. Accompanied by the strengthening or weakening of the intramolecular H-bond, the proton transfer is promoted or inhibited. A complete proton transfer is seen in MDA-BX_3_-a (X = Cl and Br) and MDA-AlX_3_-a (X = halogen), and these complexes display an equal degree of proton transfer, independent of the strength of the TrB. An electron-withdrawing group BH_2_ at the central carbon atom of MDA in BH_2_-MDA-TrH_3_-a (Tr = B and Al) can enhance the intramolecular H-bond and further cause a proton transfer. This substitution in BH_2_-MDA-BF_3_-a also strengthens the H-bond, but no proton transfer occurs.

## 5. Theoretical Methods

All the monomers and complexes were optimised using the MP2 method with an aug-cc-pVTZ basis set. Frequency calculations were performed at the same level to verify that the optimised structures were true minima on the potential energy surface, without imaginary frequencies. The interaction energies (E_int_) were calculated using supramolecular methods involving monomers with their geometries adopted in the complexes. The binding energy (E_b_) represents the difference between the energy of the complex and the sum of the monomer energies in the fully optimised structure. The difference between E_int_ and E_b_ is defined as the deformation energy (DE). These quantities were corrected for the basis set superposition error (BSSE) according to the equilibrium protocol proposed by Boys and Bernardi [42]. All the calculations were performed using the Gaussian 09 program [43].

The MEP maps of the monomers were plotted on 0.001 a.u. electron density isosurfaces at the MP2/aug-cc-pVTZ level using the Wave Function Analysis Surface Analysis Suite (WFA-SAS) software [44]. The topological parameters, including the electron density, its Laplacian, and the total energy density at the bond critical point (BCP), were calculated using the MultiWFN program [45]. Natural bond orbital (NBO) analysis was performed at the HF/aug-cc-pVTZ level to evaluate the charge transfer (CT) and interorbital interactions using the NBO 3.0 program [46]. Non-covalent interaction (NCI) [47] mapping was plotted using the Multiwfn [45] and VMD program [48]. The decomposition of the interaction energy comprised five physically significant components: the electrostatic energy (E^ele^), the exchange energy (E^ex^), the repulsive energy (E^rep^), the polarisation energy (E^pol^), and the dispersive energy (E^disp^). These features were determined at the MP2/aug-cc-pVTZ level using the local molecular orbital energy decomposition analysis (LMO-EDA) method [49] in the GAMESS program [50].

## Figures and Tables

**Figure 1 molecules-27-06091-f001:**
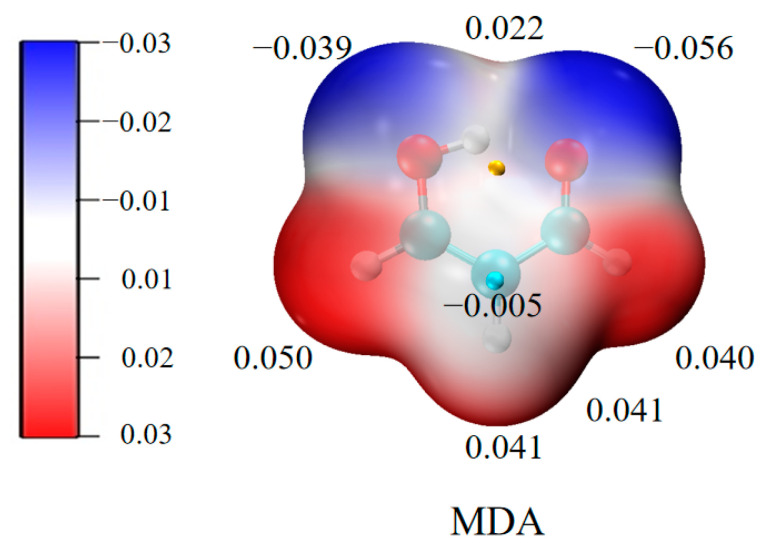
MEP maps of MDA. All are in a.u.

**Figure 2 molecules-27-06091-f002:**
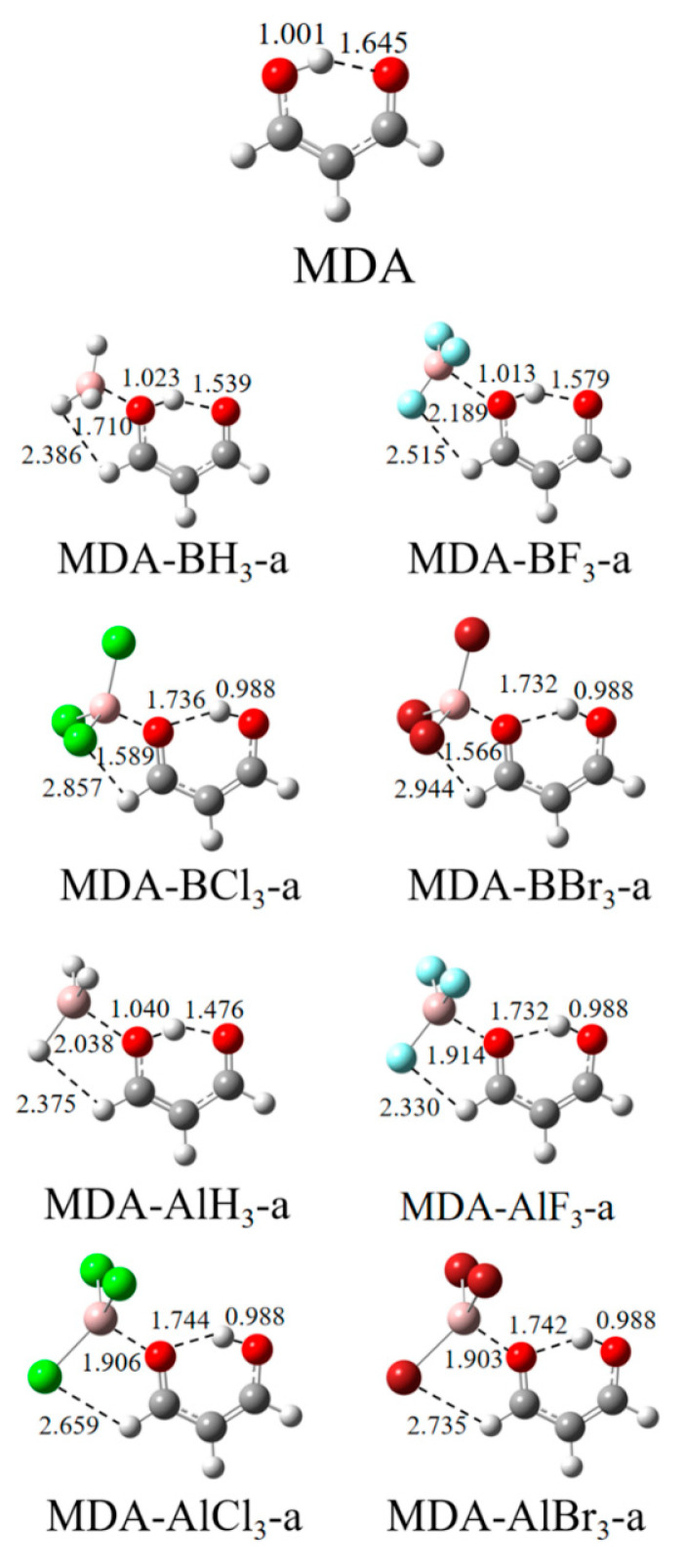
Structures of MDA and its coplanar complexes formed by the hydroxyl O with TrX_3_. Distances are in Å.

**Figure 3 molecules-27-06091-f003:**
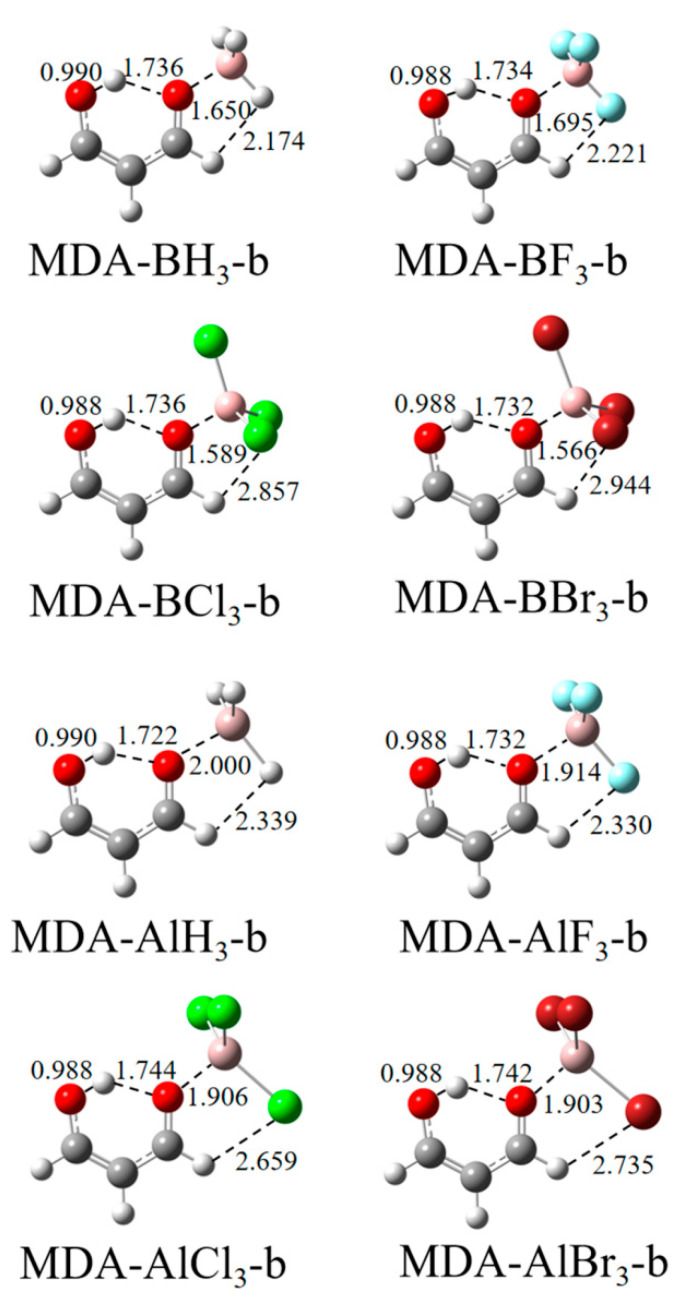
Structures of coplanar complexes of MDA formed by the carbonyl O with TrX_3_. Distances are in Å.

**Figure 4 molecules-27-06091-f004:**
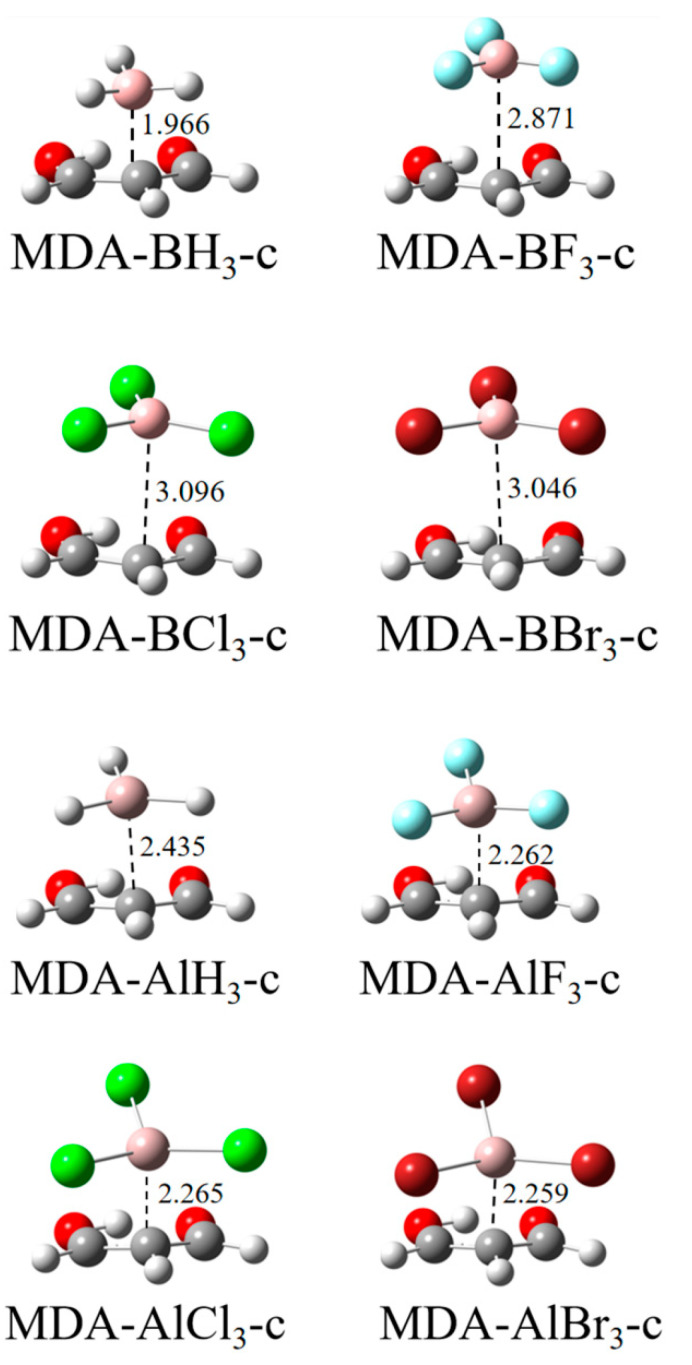
The π–π structures formed by the central carbon of MDA with TrX_3_. Distances are in Å.

**Figure 5 molecules-27-06091-f005:**
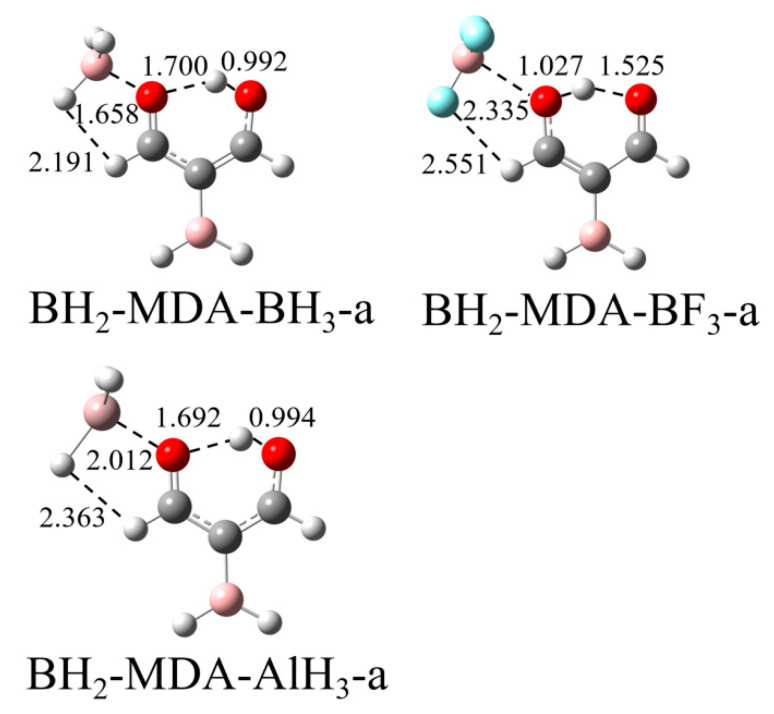
Structures of three a-type complexes substituted by a BH_2_ group.

**Figure 6 molecules-27-06091-f006:**
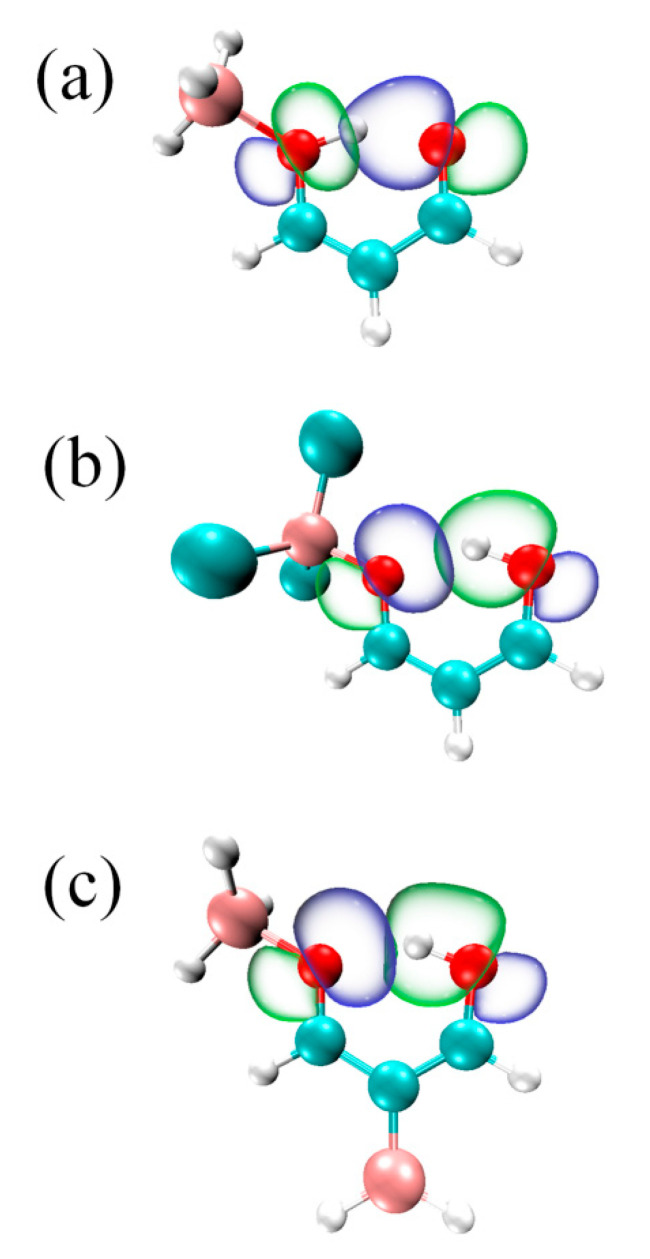
Diagrams of LpO→σ*H-O orbital interaction in (**a**) MDA-BH_3_-a, (**b**) MDA-BCl_3_-a, and (**c**) BH_2_-MDA-BH_3_-a.

**Figure 7 molecules-27-06091-f007:**
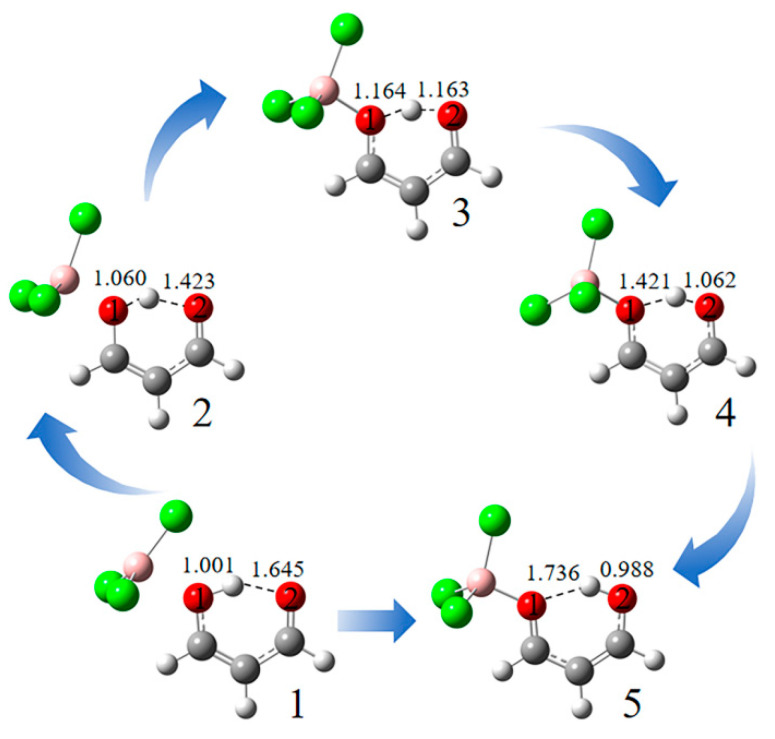
Different conformations in the formation process of MDA-BCl_3_-a.

**Table 1 molecules-27-06091-t001:** Interaction energy (E_int_), binding energy (E_b_), and deformation energy (DE) of triel bond in the complexes, all in kcal/mol.

	E_int_	E_b_	DE
MDA-BH_3_-a	−19.90	−10.83	9.07
MDA-BF_3_-a	−7.49	−5.00	2.49
MDA-BCl_3_-a	−37.36	−12.75	24.61
MDA-BBr_3_-a	−38.10	−13.94	24.16
MDA-AlH_3_-a	−20.98	−16.62	4.36
MDA-AlF_3_-a	−41.59	−32.78	8.81
MDA-AlCl_3_-a	−41.16	−31.83	9.33
MDA-AlBr_3_-a	−39.86	−30.92	8.94
MDA-BH_3_-b	−28.65	−17.60	11.05
MDA-BF_3_-b	−29.27	−10.43	18.84
MDA-BCl_3_-b	−37.36	−12.75	24.61
MDA-BBr_3_-b	−38.10	−13.94	24.16
MDA-AlH_3_-b	−26.97	−22.32	4.65
MDA-AlF_3_-b	−41.59	−32.79	8.80
MDA-AlCl_3_-b	−41.16	−31.83	9.33
MDA-AlBr_3_-b	−39.86	−30.92	8.94
MDA-BH_3_-c	−18.31	−9.68	8.63
MDA-BF_3_-c	−2.79	−2.54	0.25
MDA-BCl_3_-c	−4.12	−4.00	0.12
MDA-BBr_3_-c	−4.40	−4.20	0.20
MDA-AlH_3_-c	−11.97	−9.61	2.36
MDA-AlF_3_-c	−21.26	−14.29	6.97
MDA-AlCl_3_-c	−23.55	−14.73	8.82
MDA-AlBr_3_-c	−23.12	−14.34	8.78

**Table 2 molecules-27-06091-t002:** Electron density (ρ), its Laplacian (∇^2^ρ), energy density (H), kinetic energy density (D), and potential energy density (V) at the B∙∙∙O/C BCPs in the complexes, all in a.u.

	ρ	∇^2^ρ	H	G	V	׀V׀/G
MDA-BH_3_-a	0.0638	0.4829	−0.0210	0.1037	−0.1359	1.3105
MDA-BF_3_-a	0.0249	0.0732	−0.0028	0.0212	−0.0240	1.1321
MDA-BCl_3_-a	0.1100	0.5153	−0.0652	0.1940	−0.2591	1.3356
MDA-BBr_3_-a	0.1175	0.5575	−0.0716	0.2110	−0.2827	1.3398
MDA-AlH_3_-a	0.0382	0.2914	0.0086	0.0643	−0.0558	0.8678
MDA-AlF_3_-a	0.0575	0.4536	0.0072	0.1062	−0.0989	0.9313
MDA-AlCl_3_-a	0.0593	0.4638	0.0064	0.1095	−0.1031	0.9416
MDA-AlBr_3_-a	0.0600	0.4689	0.0062	0.1110	−0.1048	0.9441
MDA-BH_3_-b	0.0788	0.5965	−0.0280	0.1771	−0.2051	1.1581
MDA-BF_3_-b	0.0793	0.3505	−0.0416	0.1292	−0.1708	1.3220
MDA-BCl_3_-b	0.1100	0.5153	−0.0652	0.1940	−0.2591	1.3356
MDA-BBr_3_-b	0.1175	0.5575	−0.0716	0.2110	−0.2827	1.3398
MDA-AlH_3_-b	0.0447	0.3366	0.0079	0.0762	−0.0683	0.8963
MDA-AlF_3_-b	0.0575	0.4536	0.0072	0.1062	−0.0989	0.9313
MDA-AlCl_3_-b	0.0593	0.4638	0.0064	0.1095	−0.1031	0.9416
MDA-AlBr_3_-b	0.0600	0.4689	0.0062	0.1110	−0.1048	0.9441
MDA-BH_3_-c	0.0600	0.0340	−0.0377	0.0472	−0.0851	1.8030
MDA-BF_3_-c	0.0103	0.0278	0.0007	0.0064	−0.0058	0.9063
MDA-BCl_3_-c	0.0085	0.0224	0.0008	0.0049	−0.0041	0.8367
MDA-BBr_3_-c	0.0096	0.0242	0.0007	0.0054	−0.0047	0.8704
MDA-AlH_3_-c	0.0254	0.0828	−0.0023	0.0225	−0.0250	1.1111
MDA-AlF_3_-c	0.0374	0.1388	−0.0048	0.0385	−0.0436	1.1325
MDA-AlCl_3_-c	0.0394	0.1335	−0.0064	0.0386	−0.0453	1.1736
MDA-AlBr_3_-c	0.0404	0.1343	−0.0069	0.0402	−0.0471	1.1716

**Table 3 molecules-27-06091-t003:** H∙∙∙O distance (R_1_) and O-H bond length (R_2_) in the complexes as well as their difference (ΔR) relative to the monomer, all in Å.

	R_1_	ΔR_1_	R_2_	ΔR_2_
MDA-BH_3_-a	1.539	−0.106	1.023	0.022
MDA-BF_3_-a	1.579	−0.066	1.013	0.012
MDA-BCl_3_-a	0.988	−0.657	1.736	0.735
MDA-BBr_3_-a	0.988	−0.657	1.732	0.731
MDA-AlH_3_-a	1.476	−0.169	1.040	0.039
MDA-AlF_3_-a	0.988	−0.657	1.732	0.731
MDA-AlCl_3_-a	0.988	−0.657	1.744	0.743
MDA-AlBr_3_-a	0.988	−0.657	1.742	0.741
MDA-BH_3_-b	1.736	0.091	0.988	−0.013
MDA-BF_3_-b	1.734	0.089	0.988	−0.013
MDA-BCl_3_-b	1.736	0.091	0.988	−0.013
MDA-BBr_3_-b	1.732	0.087	0.989	−0.012
MDA-AlH_3_-b	1.722	0.077	0.990	−0.011
MDA-AlF_3_-b	1.732	0.087	0.988	−0.013
MDA-AlCl_3_-b	1.744	0.099	0.988	−0.013
MDA-AlBr_3_-b	1.742	0.097	0.988	−0.013
MDA-BH_3_-c	1.646	0.001	1.005	0.004
MDA-BF_3_-c	1.646	0.001	1.002	0.001
MDA-BCl_3_-c	1.642	−0.003	1.002	0.001
MDA-BBr_3_-c	1.640	−0.005	1.003	0.002
MDA-AlH_3_-c	1.639	−0.006	1.005	0.004
MDA-AlF_3_-c	1.609	−0.036	1.012	0.011
MDA-AlCl_3_-c	1.594	−0.051	1.015	0.014
MDA-AlBr_3_-c	1.593	−0.052	1.016	0.015
BH_2_-MDA-BH_3_-a	0.992	−0.653	1.700	0.699
BH_2_-MDA-BF_3_-a	1.525	−0.120	1.027	0.026
BH_2_-MDA-AlH_3_-a	0.994	−0.651	1.692	0.691

**Table 4 molecules-27-06091-t004:** Electron density (ρ), Laplacians (∇^2^ρ), and energy density (H) at the H∙∙∙O and O-H BCPs in the complexes, all in a.u.

	H···O(2)			O(1)-H		
	ρ	∇^2^ρ	H	ρ	∇^2^ρ	H
MDA	0.0533	0.1352	−0.0131	0.3220	−2.5486	−0.6945
MDA-BH_3_-a	0.0690	0.1369	−0.0241	0.2941	−2.2981	−0.6328
MDA-BF_3_-a	0.0626	0.1380	−0.0193	0.3058	−2.4059	−0.6595
MDA-BCl_3_-a	0.3364	−2.7220	−0.7339	0.0405	0.1366	−0.0044
MDA-BBr_3_-a	0.3355	−2.7156	−0.7320	0.0409	0.1382	−0.0045
MDA-AlH_3_-a	0.0810	0.1237	−0.0352	0.2766	−2.0763	−0.5819
MDA-AlF_3_-a	0.3368	−2.7123	−0.7326	0.0416	0.1338	−0.0054
MDA-AlCl_3_-a	0.3375	−2.7200	−0.7342	0.0404	0.1323	−0.0048
MDA-AlBr_3_-a	0.3373	−2.7182	−0.7337	0.0406	0.1329	−0.0048
MDA-BH_3_-b	0.0413	0.1325	−0.0053	0.3374	−2.7145	−0.7336
MDA-BF_3_-b	0.0412	0.1339	−0.0051	0.3372	−2.7180	−0.7341
MDA-BCl_3_-b	0.0405	0.1366	−0.0044	0.3364	−2.7220	−0.7339
MDA-BBr_3_-b	0.0409	0.1382	−0.0045	0.3355	−2.7156	−0.7320
MDA-AlH_3_-b	0.0429	0.1334	−0.0063	0.3353	−2.6940	−0.7286
MDA-AlF_3_-b	0.0416	0.1338	−0.0054	0.3368	−2.7123	−0.7326
MDA-AlCl_3_-b	0.0404	0.1323	−0.0048	0.3375	−2.7200	−0.7342
MDA-AlBr_3_-b	0.0406	0.1329	−0.0048	0.3373	−2.7182	−0.7337
MDA-BH_3_-c	0.0531	0.1325	−0.0131	0.3176	−2.5245	−0.6866
MDA-BF_3_-c	0.0531	0.1350	−0.0129	0.3213	−2.5464	−0.6935
MDA-BCl_3_-c	0.0537	0.1353	−0.0133	0.3210	−2.5393	−0.6921
MDA-BBr_3_-c	0.0539	0.1353	−0.0135	0.3207	−2.5358	−0.6913
MDA-AlH_3_-c	0.0540	0.1342	−0.0135	0.3170	−2.5186	−0.6854
MDA-AlF_3_-c	0.0581	0.1350	−0.0163	0.3096	−2.4522	−0.6686
MDA-AlCl_3_-c	0.0603	0.1351	−0.0179	0.3059	−2.4130	−0.6590
MDA-AlBr_3_-c	0.0605	0.1349	−0.0181	0.3053	−2.4073	−0.6575
BH_2_-MDA-BH_3_-a	0.3319	−2.6740	−0.7228	0.0452	0.1362	−0.0752
BH_2_-MDA-BF_3_-a	0.0719	0.1342	−0.0267	0.2920	−2.2490	−0.6220
BH_2_-MDA-AlH_3_-a	0.3303	−2.6560	−0.7185	0.0463	0.1362	−0.0083

## Data Availability

Not applicable.

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
