# Peer review of "Triel Bond Formed by Malondialdehyde and Its Influence on the Intramolecular H-Bond and Proton Transfer"

_molecules, 2022, doi:10.3390/molecules27186091_

Round 1

Reviewer 1 Report

Herein, coordination of the resonance assisted hydrogen bond synthon (see its role in synthetic chemistry at Chem. Eur. J. 2016, 22, 16356) of malondialdehyde to triel atom (B and Al) was studied. It was found that the carbonyl O forms a stronger triel bond than the hydroxyl O, and both of them are a better electron donor than the central carbon atom. The research wrok well done and discussed. I recommend its publication in Molecules.

Author Response

thanks

Reviewer 2 Report

The manuscript “Triel Bond Formed by Malondialdehyde and Its Influence on 3 the Intramolecular H-Bond and Proton Transfer” is devoted to the investigation of mutual influence of intramolecular hydrogen bond and intermolecular triel bond in complexes of malondialdehyde with TrX3 (Tr = B, Al; X = H, F, Cl, Br). The manuscript is well written, the computational protocol and its description are quite clear.

However, there are some points, that should be addressed by authors.

·         How many conformers of each complex were calculated? If more than one, how was the most stable/probable conformer determined?

·         It is necessary to explain the following observation: when the F atom is replaced by Cl, Br the absolute value of Eint increases for complexes with Al and decreases for complexes with B.

·         Is it possible to estimate the heights of energetic barriers along the transformations shown in Figure 7?

·         The phrase on page 6 "Thus it should be cautious to estimate the nature of TrB according to the topological parameters." requires further argumentation. Are authors sure that the observed "unexpected" positive sign of H for some complexes is due to problems of AIM theory in general? Perhaps it is due to something else, such as some artifacts for the chosen computational level or the CP search algorithm used.

·         In addition to the previous comment: analysis of other electron density parameters in BCPs (such as G and V) can also be very useful in revealing the 'nature' of the Tr...O interaction.

The list of minor comments:

·         In "Theoretical Methods": "isodensity surfaces" is better replaced by "electron density isosurfaces".

·         In "Theoretical Methods" line 98: “Laplacion” should be changed into “Laplacian”.

Author Response

The manuscript “Triel Bond Formed by Malondialdehyde and Its Influence on 3 the Intramolecular H-Bond and Proton Transfer” is devoted to the investigation of mutual influence of intramolecular hydrogen bond and intermolecular triel bond in complexes of malondialdehyde with TrX3 (Tr = B, Al; X = H, F, Cl, Br). The manuscript is well written, the computational protocol and its description are quite clear. However, there are some points, that should be addressed by authors.

Comment 1: How many conformers of each complex were calculated? If more than one, how was the most stable/probable conformer determined?

Response: For each a-type or b-type complex, there are at least two conformers with the hydroxyl O atom or carbonyl O atom to bind with one or two X atoms of TrX3, respectively. Both conformers are designed in the initial optimization, but only the former conformation is obtained for MDA-BX3 (X = Cl and Br) and only the latter conformation is obtained for the other complexes. Thus the conformation shown in Figures 2 and 3 is more stable for each complex.

Comment 2: It is necessary to explain the following observation: when the F atom is replaced by Cl, Br the absolute value of Eint increases for complexes with Al and decreases for complexes with B.

Response: This explanation has been given: This inconsistent change is mainly attributed the distortion of TrX3. BF3 shows a smaller distortion relative to BCl3 and BBr3 in the complex, thus an abnormal change occurs for the BF3 complexes.  

Comment 3: Is it possible to estimate the heights of energetic barriers along the transformations shown in Figure 7?

Response: Most structures except 5 in Figure 7 are not optimized and they are only middle structures of MDA-BCl3-a in the optimization. Thus according to these structures, the energetic barrier is not estimate.

Comment 4: The phrase on page 6 "Thus it should be cautious to estimate the nature of TrB according to the topological parameters." requires further argumentation. Are authors sure that the observed "unexpected" positive sign of H for some complexes is due to problems of AIM theory in general? Perhaps it is due to something else, such as some artifacts for the chosen computational level or the CP search algorithm used.

Response: The energy density is also positive for the Al∙∙∙N BCP in pyrazine∙∙∙AlX3 (X = H, F, Cl, Br, and CH3) with the interaction energy larger than 26 kcal/mol at the MP2/aug-cc-pVDZ level [ChemPhysChem, 2018, 19, 3122-3133]. The similar result is also found in Ref. [ChemPhysChem 2015, 16, 1470–1479]. When we applied the M06-2X/aug-cc-pVTZ method to calculate the AIM parameters of MDA-AlX3, the sign of energy density is still positive. Thus our statement is reliable.

Comment 5: In addition to the previous comment: analysis of other electron density parameters in BCPs (such as G and V) can also be very useful in revealing the 'nature' of the Tr...O interaction.

Response: This analysis has been performed, the related data has been added in Table 2, and the following sentences have been added: which is also confirmed by the magnitude of ׀V׀/G (>1), and the magnitude of ׀V׀/G is smaller than 1

Comment 6: In "Theoretical Methods": "isodensity surfaces" is better replaced by "electron density isosurfaces".

Response: This replacement has been done as suggested by the reviewer.

Comment 7:  In "Theoretical Methods" line 98: “Laplacion” should be changed into “Laplacian”.

Response: This error has been corrected, thanks

Reviewer 3 Report

Very interesting work, authors have presented a detailed work with electronics and conformational analysis of one of the key reagents in organic chemistry. The Pi-Pi interaction studies add an additional factor to the work.

page 1, line 25 disobserb
page 3, figure1, is it possible to add a 3d-confirmation of the molecule as it exists in the computational model.   query 1: how do you remove possibility of a counter ion, when boron compounds interact with hydroxy group. like MDA-BX3-a structures. as the compounds can have a anion species, BX4-   query 2: the BH3 would be in general dimer, did authors observe any extended color pattern   query 3: figure 2 MDA-BCl3-a and many others don't show a hydroxy interaction to the Tr salts. they show with the carbonyl interaction, the figures need revision.   query 4: can authors a interaction with BMe3 or BPh3 derivatives, as authors have mentioned these reagents are finding new uses recently as lewis acidic catalysts.        

Author Response

Very interesting work, authors have presented a detailed work with electronics and conformational analysis of one of the key reagents in organic chemistry. The Pi-Pi interaction studies add an additional factor to the work.

Comment 1: page 1, line 25 disobserb

Response: This word has been written as disorder

Comment 2: page 3, figure 1, is it possible to add a 3d-confirmation of the molecule as it exists in the computational model.

Response: This figure has been modified.

Comment 3: how do you remove possibility of a counter ion, when boron compounds interact with hydroxy group. like MDA-BX3-a structures. as the compounds can have a anion species, BX4-

Response: These complexes are neutral. If an anion species, BX4- appears as suggested by the reviewer, we think that a counter ion would occur on the hydroxyl H atom. According to the AIM data at the Tr∙∙∙O BCP, we think no such anion species is present since the Tr∙∙∙O interaction is a partially covalent interaction not a covalent bond.

Comment 4: the BH3 would be in general dimer, did authors observe any extended color pattern.

Response: Yes, BH3 would be in general dimer. The B atom has a shallow σ-hole in the BH3 dimer, while this atom shows a deep π-hole in BH3. BH3 is often taken as a Lewis acid in studying the triel bond, while its dimer is seldom as a Lewis acid in studying the triel bond. The weak triel bond formed by the BH3 dimer will not cause a proton transfer in MDA, thus this dimer has not been studied.

Comment 5: figure 2 MDA-BCl3-a and many others don't show a hydroxy interaction to the Tr salts. they show with the carbonyl interaction, the figures need revision.

Response: In some complexes of figure 2, the triel atom binds with the hydroxyl O atom at the initial stage, but this hydroxyl O atom changes into the carbonyl O atom in the end due to the proton transfer. Therefore these figures are right.

Comment 6: can authors a interaction with BMe3 or BPh3 derivatives, as authors have mentioned these reagents are finding new uses recently as Lewis acidic catalysts.

Response: This suggestion is good. However, the B atom has a shallower π-hole in BMe3 than that in BH3 [Appl. Organometal Chem. 2018, 32, e4367], thus the former forms a weaker triel bond and does not cause a proton transfer in MDA. This similar result will also be found for BPh3 since there is delocalization between the empty p orbital on the B atom and the π electrons on the benzene ring, also resulting in a shallower π-hole. Thus these calculations are not performed.